# Bovine Serum Albumin-Based Nanoparticles: Preparation, Characterization, and Antioxidant Activity Enhancement of Three Main Curcuminoids from *Curcuma longa*

**DOI:** 10.3390/molecules27092758

**Published:** 2022-04-25

**Authors:** Andrea Mariela Araya-Sibaja, Krissia Wilhelm-Romero, María Isabel Quirós-Fallas, Luis Felipe Vargas Huertas, José Roberto Vega-Baudrit, Mirtha Navarro-Hoyos

**Affiliations:** 1Laboratorio Nacional de Nanotecnología LANOTEC-CeNAT-CONARE, Pavas, San José 1174-1200, Costa Rica; krissia.wilhelm@ucr.ac.cr (K.W.-R.); jvegab@gmail.com (J.R.V.-B.); 2Laboratorio BIODESS, Escuela de Química, Universidad de Costa Rica, San Pedro de Montes de Oca, San José 2060, Costa Rica; maria.quirosfallas@ucr.ac.cr (M.I.Q.-F.); luis.vargashuertas@ucr.ac.cr (L.F.V.H.); mnavarro@codeti.org (M.N.-H.); 3Laboratorio de Investigación y Tecnología de Polímeros POLIUNA, Escuela de Química, Universidad Nacional de Costa Rica, Heredia 86-3000, Costa Rica

**Keywords:** nanoparticles, bovine serum albumin, curcumin, desmethoxycurcumin, bisdemethoxycurcumin, characterization, antioxidant activity

## Abstract

Bovine Serum Albumin (BSA) lipid hybrid nanoparticles are part of the new solutions to overcome low bioavailability of poor solubility drugs such as curcuminoids, which possess multiple biological advantages; however, they are counterbalanced by its short biological half-life. In this line, we prepared the three main curcuminoids: curcumin (CUR), desmethoxycurcumin (DMC), and bisdemethoxycurcumin (BDM)-loaded BSA nanoparticles. The three formulations were characterized by the average size, size distribution, crystallinity, weight loss, drug release, kinetic mechanism, and antioxidant activity. The developed method produced CUR-, DMC-, and BDM-loaded BSA nanoparticles with a size average of 15.83 ± 0.18, 17.29 ± 3.34, and 15.14 ± 0.14 nm for CUR, DMC, and BDM loaded BSA, respectively. FT-IR analysis confirmed the encapsulation, and TEM images showed their spherical shape. The three formulations achieved encapsulation efficiency upper to 96% and an exhibited significantly increased release from the nanoparticle compared to free compounds in water. The antioxidant activity was enhanced as well, in agreement with the improvement in water release, obtaining IC_50_ values of 9.28, 11.70, and 15.19 µg/mL for CUR, DMC, and BDM loaded BSA nanoparticles, respectively, while free curcuminoids exhibited considerably lower antioxidant values in aqueous solution. Hence, this study shows promises for such hybrid systems, which have been ignored so far, regarding proper encapsulation, protection, and delivery of curcuminoids for the development of functional foods and pharmaceuticals.

## 1. Introduction

One of the most impactful areas for drug delivery is the production of biocompatible nanoparticles. Nanoparticle formulation has demonstrated to be an effective approach for the protection and targeted delivery of nutraceuticals, moved by the versatile properties of proteins and hybrid drug carriers fabricated from proteins and synthetic polymers, which are increasingly described in the literature [1,2,3,4]. In particular, the albumin protein presents a long range of capabilities, including the nontoxic properties, the high biodegradation metabolized in vivo to produce safe degradation products, water solubility, and non-immunogenicity [5,6]. In addition, albumin nanoparticles have different drug binding sites increasing the amount of drugs that can be loaded in the matrix [7]. This self-assembly is enhanced due to the presence of a lipid matrix, which minimizes the leakage of the encapsulated content during preparation, increasing drug-loading efficiency leading to the formation of micelles or other aggregates in aqueous media [4,8,9].

In particular, bovine serum albumin (BSA) contains two tryptophan residues, Trp 134 and Trp 212, and in the native form, these residues are located on the surface of the molecule and in an hydrophobic pocket [10]. This fact increases the affinity of polyphenols for BSA with protein unfolding amine groups, thiol groups, and hydrophobic regions relocating to the surface of the molecule and causing an increase in intra-molecular binding along with a reduction in hydrophobic interactions [7,11,12]. There are a few literature references that indicate the use of lipids with BSA [13], which can self-assemble, resulting in micellar nanoparticles and other nanoparticles with protein shell [7]. Other advantages of the incorporation of lipid in the nanoparticle matrix is the ease of large-scale production; biocompatible and biodegradable nature of the materials; low toxicity potential; possibility of controlled and modified drug release; and the solubility enhancement of poor solubility drugs [9,14,15].

In this regard, curcuminoids have been extensively reported to possess multiple advantages. The three main curcuminoids in *Curcuma longa* are curcumin (CUR), demethoxycurcumin (DMC), and bisdemethoxycurcumin (BDM). In the last years, many of in vitro and in vivo experiments for the three curcuminoids have been performed, from cell cultures to clinical trials [16,17,18,19,20]. The similarity in their structures suggests that they may exhibit similar bioactivities, and it is reported that they may have potential as chemo-preventive and therapeutic agents in inflammatory disorders and cancer [21]. However, these properties are counterbalanced by its short biological half-life and poor solubility resulting in poor absorption, and low bioavailability via the oral route, which is the main limitation of applicability of curcuminoids as drugs [22]. Therefore, the use of hybrid systems based on BSA can help to overcome these limitations.

In this study the elaboration, the characterization and in vitro antioxidant evaluation of three BSA-based nanoparticles containing CUR, DMC, and BDM are reported. The molecular structures of curcuminoids studied are illustrated in Figure 1.

## 2. Results

### 2.1. Characterization of CUR, DMC, and BDM Nanoparticles

#### 2.1.1. Encapsulation

The FT-IR technique helps to determine the interactions in the encapsulation through the probable bonding between the drug and carrier [23]. The FT-IR spectra were evaluated between 400 and 4000 cm^−1^, and the results are shown in Figure 2. The FT-IR spectra of the components of nanoparticles show plain NP, free CUR, DMC, and BDM as well as the curcuminoid-loaded NP. The FT-IR spectra of BSA showed characteristic peaks at 3447.0, 2918.2, 2849.6, 2103.3, 1636.4, 1459.5, 1292.8, and 716.6 cm^−1^ due to O–H stretching and hydrogen bonding, C–H stretch, –C=C– alkynes stretch, N–H bonding of primary amines, C–C aromatic stretch and C–H bond, N–O symmetric stretch, and C–H bending vibrations, respectively. The main peaks for the curcuminoids correspond to 3363 cm^−1^ related to the stretching vibration of hydrogen-bonded (-OH), 1649 cm^−1^ of the C=O stretching vibration, and 1270 cm^−1^ due to C-O stretching.

As shown in Figure 2, all the characteristic absorption peaks of BSA and the curcuminoids can be found in the FT-IR spectra of the CUR-, DMC-, and BDM-loaded NP, which indicated a combination of these signals from the curcuminoids as well as from BSA and Chol components, in which FT-IR is a multifunctional and powerful tool to monitor the changes of functional groups in biopolymers and curcuminoids [24]. For instance, signals at 1463 cm^−1^, 1054 cm^−1^, and 742 cm^−1^ correspond to CH_2_ and CH_3_ deformation vibrations, ring deformation, and C-H out-of-plane bending from Chol [25]. In addition, the FT-IR spectra of CUR-NP-, DMC-NP-, and BDM-NP-loaded and the plain NP share peaks at wavenumbers corresponding to 3318 cm^−1^; to the stretching vibration of hydrogen-bonded (-OH) 2928 cm^−1^ due to the stretching vibrations of methyl (CH_3_); to 2846 cm^−1^ corresponding to C-H stretch aliphatic; 1707 cm^−1^ C-H-C stretching vibration; 1370 cm^−1^ in-plane O-H bend; 1111 cm^−1^ C-O stretch; and 959 cm^−1^ and 797 cm^−1^ to stretching of C-H for the DMC. The signals at 2846 cm^−1^ correspond to C-H stretch aliphatic; 1707 cm^−1^ C-H-C stretching vibration; 1347 cm^−1^ in-plane O-H bend; 1103 cm^−1^ C-O stretch; and 938 cm^−1^ and 761 cm^−1^ relative to the stretching of C-H for the CUR and BDM as hybrid NPs further demonstrating that the drug was loaded onto the nanoparticle core. As shown in the results, the FT-IR spectra of the plain NP and the curcuminoid loaded NP did not present significant differences between the bands, which is indicative of the successful loading of curcuminoids onto the nanoparticles core, thus being absent on the nanoparticle’s surface.

#### 2.1.2. Particle Size, Particle Size Distribution, Morphology, and Encapsulation Efficiency

Particle size, polydispersity, and encapsulation efficiency (%EE) are important parameters for quality control of nanoparticles. The size average, Polydispersity Index (PDI), encapsulation efficiency (%EE), loading capacity (%LC), and yield of CUR, DMC, and BDM loaded NP are shown in Table 1.

When comparing the sizes obtained by DLS for the three formulations, it was observed that the nanoparticles had small sizes, and these were found to be between 15 and 17 nm for the nanoparticles loaded with curcuminoids. Showing slight differences, BDM-NP presented the smallest size, followed by CUR-NP, and finally DMC-NP. In addition, plain NP exhibited a size of around 20 nm; therefore, no significant difference between plain and curcuminoid-loaded NP was observed. Particle size, polydispersity, and %EE are important parameters to help determine stability and the loaded function of the nanoparticles due to the influence in the release of the compound inside the nanoparticle.

The PDI, on the other hand, is a dimensionless measure of the breadth of the particle size distribution [26,27]. The results show that the smallest value for PDI was for BDM-NP, CUR-NP as second, and DMC-NP was in the third place. The PDIs of BDM-NP and CUR-NP formulations were below 0.3, which indicated that the particle size exhibited narrow distribution and DMC-NP presented a value slightly above 0.3, which was on the edge of being considered to have a broad distribution [28,29,30].

Concerning the % EE of CUR, DMC, and BDM, the three curcuminoids were efficiently loaded into the BSA-based NP, achieving an encapsulation efficiency <96%. Therefore, a high EE meant that the curcuminoid maximal solubility in the lipid was reached in the NP and that all molecules remained in the particles after lipid solidification [31], and it is expected to be higher than 90%, as previous reports have stated in the literature [32,33]. The high values of EE can be attributed to phenyl groups on the curcuminoids structure loaded into BSA. The results show no significant difference in encapsulation efficiency between CUR-NP and DMC NP-BSA; however, a significant difference was observed between these and BDM NP-BSA (*p* < 0.05).

HR-TEM and size distribution histogram obtained by conducting DLS measurements of the three curcuminoid-loaded nanoparticle formulations are shown in Figure 1. Comparing the three formulations, it was observed that the curcuminoids loaded in the BSA-NP shown in Figure 3 present a spherical shape with a porous morphology. Most particles were observed to be distributed between 50 and 20 nm under HR-TEM with almost similar dimensions of CUR-NP and DMC-NP observed in DLS data. This difference in size observed between HR-TEM and DLS data in BDM NP-BSA may be due to the of the differences between the dynamic diameter observed in DLS and the size in TEM since the former is a diameter calculated from the diffusional properties of the particle and indicates the real size of the hydrated and solvated dynamic particle [34].

A formulation composed of BSA, Chol, 1-ethyl-3-(3-dimethylaminopropyl) carbodiimide (EDC), and CUR reports a CUR-loaded NPs size of 92.59 nm [35], while Amano et al. [36] reported a liposomal nanoformulation composed of CUR, BSA, or human serum albumin with a size of 71.4 ± 9.0 nm. There are no reports of BSA nanoformulation of DMC and BDM to our knowledge; however, the liposomal nanoformulation of BDM with cholesterol with an average size of 110.8 ± 8.5 nm has been reported [32]. Our findings indicate smaller particle sizes and synthesized BSA nanoparticles for other curcuminoids such as DMC and BDM.

#### 2.1.3. Differential Scanning Calorimetry (DSC)

DSC thermograms of lyophilized plain NP, CUR-NP, DMC-NP, and BDM-NP are presented in Figure 4. Free CUR, DMC, and BDM exhibit melting temperatures at 184 °C, 172 °C, and 222 °C, respectively [37]. The curcuminoid-loaded NPs did not present endothermic events related to their melting temperatures. Plain NP was characterized by the existence of an endothermic peak at 119.90 °C; meanwhile, CUR-NP and DMC-NP showed sharp peaks at 125.76 °C and 135.61 °C, respectively. BDM-NP presented an exothermic event at 101.53 °C. All these events can be associated with polymorphic forms and phase transitions reported for Chlo [38] that could be occurring during NP preparation or lyophilization processes. The absence of the characteristic endothermic peaks of CUR, DMC, and BDM suggested that the curcuminoids were molecular binding and were molecularly dispersed within the nanoparticle matrix [39,40]. It is commonly preferred that the drug in the formulation is amorphous, resulting in better dissolution, absorption, and bioavailability [35]. Moreover, there was no evidence of incompatibilities between curcuminoids and formulation constituents.

#### 2.1.4. Thermogravimetric Analysis (TGA)

Figure 5 shows the TGA curves of the plain-NP, CUR-NP, DMC-NP, and BDM-NP. Similar thermal behaviors were observed for the three curcuminoids hybrid nanoparticles and the plain NP. Three mass loss events at 92.3 °C were observed that can be associated with water loss at around 300 °C and 405 °C. Only a slight difference in the thermal decomposition of CUR-NP was observed when starting its weight loss at a slightly lower temperature than DMC-NP and BDM-NP.

According to Hefferman et al. (2017), free CUR showed a weight loss at around 184 °C, DMC at 172 °C, and BDM at 142 °C [41]. Their results suggested that BDM is less thermal stable than CUR and DMC. In this scenario, the developed BSA-based hybrid NP containing curcuminoids exhibited an increase in thermal stability compared with the pure curcuminoids in alignment with thermal stability observed by Cho [42] and BSA [43].

### 2.2. In Vitro Evaluation of CUR, DMC, and BDM Hybrid Nanoparticles

#### 2.2.1. In Vitro Curcuminoids Release Studies

The curcuminoids hybrid nanoparticles were studied for the release of CUR, DMC, and BDM in two different dissolution media M_1_ (phosphate buffered pH 6.8 with 20% of MeOH) and M_2_ (water). Although water is not recommended as a release medium for poorly water-soluble molecules, it was also tested in an attempt to evaluate the aqueous solubility enhancement exerted by the synthesized nanosystems. However, for compounds exhibiting poor water solubility, the use of dissolution media-containing surfactants or organic solvents is allowed. Figure 6 shows the release profile of the curcuminoids from the hybrid nanoparticles in both dissolution media over the period of 120 min. The curcuminoids release in M_1_ (pH 7.4) appeared similar for DMC and BDM in both media, whereas it showed relatively faster release with the release of a higher amount of CUR as compared to that in M_2_ (water, pH 6.8). The presence of MeOH in M_1_ suggested an increased dissolution rate of free curcuminoids and a higher release rate of curcuminoids from the nanoparticle.

Nevertheless, comparing the dissolution profile of free CUR, DMC, and BDM with curcuminoids release from BSA-based NP in water, observed in Figure 7, the releases increased. At 120 min, only 5.5% of free BDM dissolved, while its release from the hybrid nanosystem was 55.0% at 120 min. In turn, only 2.8% of free DMCs dissolved after 120 min, while 25.0% were released from the nanoparticle at 120 min; for CUR at 120 min, only 6.8% of free curcumin was dissolved, while its release from the hybrid nanosystem was 35.0% at 120 min. The release of a loaded drug molecule from the shell-core largely depends on hydrophobic interactions between the inner core and drug, as previously mentioned. The increased release of DMC and BDM from the hybrid nanoparticles can be attributed to the hydrophobic interaction between the curcuminoid and the bilayer; as the hydrophobic interaction becomes weak, the shell core breaks and exhibits a fast and sustained release of the molecule [44].

The release behavior of the curcuminoids from the nanoparticles is thought to be dependent on bind with the BSA inner core. The BSA molecule has many free carboxyl groups, which can bind positively charged curcuminoid. Therefore, the curcuminoid could be easily loaded into the BSA-based NP with high efficiency, as we previously reported. Since the curcuminoid is mainly loaded through electrostatic attraction with carboxylic groups of BSA, this interaction is influenced by pH, for which its degree of charge is related to the dissolution media. The diverse factors contributing to an efficient release include a large surface area, a high diffusion coefficient due to small molecular size, low viscosity in the matrix, and a short diffusion distance δ for the drug (i.e., release from the outer surface region of the nanoparticle) [45,46,47].

#### 2.2.2. Kinetic Parameters of Drug Release

Table 2 lists the kinetic parameters of curcuminoids release from CUR-NP, DMC-NP, and BDM-NP in M_1_ and M_2_ obtained after fitting the data to six mathematical model equations shown in the Experimental Section in order to understand transport mechanisms. Appendix A show the best fitting models for the three curcuminoid BSA-based NPs. It is evident that the release of curcuminoids differs in the fit to all the used models. CUR-NP best fitted for zero order and first order in M_1_ and M_2_, respectively. However, R^2^ values were quite similar in both models in M_1_ for first-order DMC in M_1_ and Higuchi in M_2_, whereas BDM fitted best into the Ritger–Peppas model for both media. The statistical best fit indication can be observed by the large correlation coefficient (R^2^ < 0.950) values obtained in the case of mentioned kinetic models.

The controlled release process could be defined by the rate at which the process continues. The process rate is determined by defining its order. In the case of CUR, the first and zero-order indicate the permeability of the water of the core, which would be hydrated, and the active agent is released until reaching its solubility or saturation concentration. Then, the drug diffuses through the inner core and is released at a constant rate because of the saturation condition in the core that produces a steady concentration gradient through it. This indicates that the release rate of CUR from nanoparticles in the media is significant concentration dependent.

DMC kinetic shows reasonable fitting to the Higuchi model in M_2_, indicating that the mechanism of release is predominantly diffusional [48]; thus, DMC in M_1_ showed an acceptable fitting first, which suggested that in this medium, the release rate is directly proportional to the concentrations of drug in agreement with the presence of an organic solvent in M_1_.

BDM, on the other hand, obtained its best fit for the Ritger–Peppas model. In this model, for spherical systems, the release involves a Fickian mechanism, including molecular diffusion of the drug due to a chemical potential gradient; if 0.43 *< n <* 0.85, the release is anomalous transport release wherein both diffusion and relaxation pathways contribute to the release. In the case of BDM release, the n values were 0.5 < *n* ≤ 0.89 in both media, suggesting that anomalous transport is the predominant mechanism of the release. Moreover, the *n* values around 0.5 or upper also indicate the release mechanism in film delivery systems, and it could be the case that BDM-NP agglomerate and form films during their diffusion [49].

#### 2.2.3. Antioxidant Activity Evaluation of Free and Curcuminoids Hybrid Nanoparticles

The antioxidant activities of nanoencapsulated curcuminoids were studied by conducting DPPH analysis, as described in the respective Experimental Section. Results of the antioxidant activity are shown in Table 3.

Results for antioxidant activity evaluation of the free curcuminoids in EtOH showed CUR to have the lowest IC_50_ and, therefore, 23% and 46% higher antioxidant activity than DMC and BDM, respectively. This trend is consistent with previously reported results for the antioxidant activity of the three free curcuminoids, where the antioxidant activity of CUR was found to be higher than the activity of DMC, followed by BDM [51]. In turn, DPPH analyses of the aqueous solutions of the free curcuminoids showed considerably higher IC50 for all three metabolites and, thus, much lower antioxidant activity, which is associated with their low solubility in water. In addition, an opposite trend of antioxidant activity was shown for these aqueous samples, with CUR yielding the highest IC_50_, indicating an antioxidant activity that is 43% higher for BDM in water, which is in alignment with the relative higher aqueous solubilities for BDM with respect to CUR [18].

DPPH findings for the three BSA nanoparticles showed an important decrease in IC_50_ values; hence, a significant enhancement of antioxidant activity was observed with respect to free curcuminoids in aqueous solution. On the other hand, IC_50_ of curcuminoids in BSA nanoparticles was shown to be similar to those of free curcuminoids in ethanolic solutions, with values indicating that antioxidant activity improved by nanoencapsulation. In fact, CUR’s antioxidant activity increased by 3%, DMC’s by 6%, and BDM´s by 15%.

Comparing with the literature, the antioxidant activities of curcuminoids DMC and BDM in BSA nanoparticles show better values (8% and 14%, respectively) in comparison to those obtained for hybrid polymeric-lipid nanoparticles previously reported for these two metabolites [50]. In turn, the results obtained for the antioxidant activity of CUR in BSA nanoparticles were also consistent with the enhancement shown in previous results for CUR in other nanoparticle formulations [52,53]. Finally, the overall results for the three curcuminoids in the BSA nanoparticles aligned with improvement findings shown for the nanoparticles of curcuminoid mixtures [54].

## 3. Materials and Methods

Curcumin (CUR), demethoxycurcumin (DMC), and bisdemethoxycurcumin (BDM) were obtained and isolated from Costa Rican *Curcuma longa* from commercial cultivars as reported by BIODESS Laboratory (Costa Rica). CUR, DMC, and BDM analytical standards for quantification, cholesterol (Chol), Bovine Serum Albumin lyophilized powder (BSA), 2,2-diphenyl-1-picrylhidrazyl (DPPH), dichloromethane (CH_2_Cl_2_), phosphoric acid (H_3_PO_4_), and disodium hydrogen phosphate were purchased from Sigma–Aldrich (Saint Louis, MO, USA). Sodium dihydrogen phosphate monohydrate was acquired from Merck (Kenilworth, IL, USA). Polysorbatum 80 (Tween 80) was purchased from Sonntag & Rote S.A. (San José, Costa Rica), and sorbitan monooleate (Span^®^ 80) was supplied by LABQUIMAR S.A (San José, Costa Rica). Chloroform (CHCl_3_), methanol (MeOH), ethanol (EtOH), and acetonitrile (MeCN) were purchased from JTBaker (Phillipsburg, NJ, USA). All solvents were HPLC/UV grade or highly pure, and the water was purified using a Millipore system filtered through a Millipore membrane 0.22 µm Millipak 40.

### 3.1. Preparation of CUR, DMC, and BDM Hybrid Nanoparticles

CUR, DMC, and BDM nanoparticles were prepared according to Wilhelm et al. (2021) [50] substituting Pluronic-F127 by BSA. The method consisted of preparing an emulsion as follows: An aqueous phase was prepared by dissolving 250 mg of BSA in 50 mL of acetic acid 0.1%, and a 1:1 mixture of Tween 80: Span80. Additionally, an organic phase was prepared using 5 mg of the intended CUR, DMC, or BDM and 120 mg of Chol dissolved in 6 mL of MeOH: CHCl_3_ 1:1 solvent mixture. Then, the organic phase was added to the aqueous one at approximately 3 mL/min and mixed at 16,000 rpm for 10 min using an IKA ULTRA-TURRAX^®^ T25 high-speed homogenizer (Staufen, Germany). The nanoparticles were collected by ultracentrifugation using a Thermo Scientific Sorvall ST 16R centrifuge(Thermo Fisher Scientific, Tokyo, Japan) at 12,000 rpm for 40 min at 10 °C. Unreacted substances were removed by washing three times with ultrapure water and filtered through an ADVANTEC^®^ ultrafilter unit (Tokyo, Japan). For in vitro evaluation, the final nanoformulations were dispersed in 5 mL of purified water containing 0.01% Tween80, and for further physicochemical characterization, samples were lyophilized in a continuous freeze-dryer Buchi, Lyovapor L-300 (Tokyo, Japan).

The formulations containing CUR, DMC, or BDM were identified as CUR-NP, DMC-NP, and BDM-NP, respectively. A blank of NP was prepared following the aforementioned method without the curcuminoids in the organic phase and identified as Plain-NP.

### 3.2. Physicochemical Characterization of CUR, DMC, and BDM Hybrid Nanoparticles

#### 3.2.1. Fourier Transform Infrared Spectroscopy (FT-IR)

FT-IR spectra were collected from a Thermo Scientific Nicolet 6700 spectroscope equipped with a diamond attenuated total reflectance (ATR) accessory. The samples were placed into the ATR cell without dilution and analyzed in the range of 4000–600 cm*^−^*^1^, collecting 32 scans at a resolution of 4 cm*^−^*^1^.

#### 3.2.2. High Resolution Transmission Electron Microscopy (HR-TEM)

HR-TEM images were acquired using a JEOL, JEM2011 HR-TEM, at an acceleration voltage of 120 kV. Samples measuring 5 μL were placed on a sample holder grid and dried under a nitrogen atmosphere.

#### 3.2.3. Dynamic Light Scattering (DLS)

Particle size (PS, z-average) and polydispersity index (PI) were determined using a Malvern Nano Zetasizer ZS90 instrument (Malvern Panalytical, Almelo, The Netherlands) using medium refractive index 1.33, and viscosity was 0.8872 cP under 90°. Samples were diluted with deionized water to achieve appropriate concentrations, and measurements were performed at 25 °C.

#### 3.2.4. Encapsulation Efficiency (EE)

EE was determined using the direct method of quantifying the real amount of each curcuminoid CUR, DMC, or BDM into the nanoparticles by breaking the nanoparticles with organic solvent and quantifying the amount of CUR, DMC, and BDM in the respective formulation, in which 100 µL of fresh nanoparticles samples was dissolved in 900 µL of MeOH. The samples were filtered through a 0.45 μm polyamide membrane placed in a Sartorius stainless steel syringe filter holder. Solutions measuring 10 μL were injected in a Dionex Ultimate 3000 UHPLC system equipped with a variable wavelength detector, pump, variable temperature compartment column, and autosampler. The chromatographic elution was carried out in a Nucleosil 100-5 C18 column (250 mm × 4.0 mm, 5 μm) at a temperature of 35 °C using 55% of MeCN and 45% of H_3_PO_4_ 0.1% as mobile phase at a flow rate of 1 mL/min and setting down the detection at 420 nm. The %EE, the LC in mg/g, and % yield for each curcuminoid formulations were calculated with Equations (1)–(3), respectively.
(1)$$\%\mathrm{EE}=\frac{\mathrm{Amount}\;\mathrm{of}\;\mathrm{CUR},\;\mathrm{BDM}\;\mathrm{or}\;\mathrm{DMC}\;\mathrm{onto}\;\mathrm{nanoparticle}}{\mathrm{Total}\;\mathrm{CUR},\;\mathrm{BDM}\;\mathrm{or}\;\mathrm{DMC}\;\mathrm{added}}\times 100$$
(2)$$\mathrm{LC}\;(\frac{\mathrm{mg}}{g})=\frac{\mathrm{mg}\;\mathrm{of}\;\mathrm{encapsulated}\;\mathrm{CUR},\;\mathrm{BDM}\;\mathrm{or}\;\mathrm{DMC}}{g\;\mathrm{of}\;\mathrm{total}\;\mathrm{nanoparticles}\;\mathrm{obtained}}$$
(3)$$\%\;\mathrm{yield}=\frac{\mathrm{weigth}\;\mathrm{of}\;\mathrm{nanoparticles}\;\mathrm{obtained}}{\mathrm{weigth}\;\mathrm{of}\;\mathrm{starting}\;\mathrm{materials}}\times 100$$

#### 3.2.5. Thermal Analyses

Differential Scanning Calorimetry (DSC) curves were obtained from a TA Instruments DSC-Q200 calorimeter equipped with a TA Refrigerated Cooling System 90. Samples ranging from 2 and 5 mg each were placed in aluminum pans with lids that were kept unsealed. The measurement was conducted from 40 to 250 °C using a heating rate of 10 °C/min under a dynamic nitrogen atmosphere of 50 mL/min.

Thermogravimetric analyses were conducted on a TA Instruments model Q500 thermogravimetric analyzer. Approximately 5 mg of the sample was in place in a platinum crucible evaluated from 25 to 800 °C using a temperature ramp of 10 °C/min under a nitrogen atmosphere flow of 10 mL/min on the sample and 90 mL/min on the microbalance.

### 3.3. In Vitro studies of Curcuminoids Hybrid NP

#### 3.3.1. Release profile of CUR, DMC, and BDM from Hybrid NP

The in vitro release profile of each formulation was estimated using two different dissolution media. A phosphate buffered saline of pH 7.4 with 20% of MeOH and 2.5% of Tween 80 was used as medium 1 (M1), and water was used as medium 2 (M2). An amount of 1 mL of each formulation or 1.0 mg of the free curcuminoid was placed in 80 mL of the respective medium maintained at 37 ± 0.5 °C and 150 rpm constant agitation in a Labnet 211 DS shaking incubator. At specific time intervals, a volume of 4 mL of each solution was withdrawn, and this medium volume was not replaced. The collected solutions were centrifuged at 6000 rpm for 10 min using a Thermo Scientific Sorvall ST 16R centrifuge at a controlled temperature of 37 °C. The concentrations of CUR, DMC, and BDM were determined using a double beam spectrophotometer Shimadzu 1800 UV-Vis at a wavelength of 420 nm. The dissolution profiles of pure CUR, DMC, and DMC were evaluated in the two media for comparison with curcuminoid release from the developed nanoparticles. The sampling was performed in triplicate.

#### 3.3.2. Release Kinetic Models

To understand the probable mechanism of the curcuminoids release from the nanoparticles in M_1_ and M_2_, six different mathematical models were used. All mathematical models require the fitting of experimental data of the cumulative amount of curcuminoid released at time [*A*]*_t_* and at equilibrium time (*t*) versus time to the empirical equation of the corresponding mathematical models shown below:(i)The zero-order drug delivery model is expressed by the following equation [49].(4)$${\left[A\right]}_{t}=-\mathrm{kt}+{\left[A\right]}_{0}$$
(ii)The first-order drug delivery model is expressed by the following equation [55]:(5)$$\ln{\left[A\right]}_{t}=-\mathrm{kt}+\ln{\left[A\right]}_{0}$$
where [*A*]*_t_* represents the % of drug release, k is the first-order release model constant, and [*A*]_0_ is the intersection with the axis.(iii)The Second-order drug delivery model is expressed by the following equation.(6)$$\frac{1}{{\left[A\right]}_{t}}=\mathrm{kt}+\frac{1}{{\left[A\right]}_{0}}\;$$
(iv)The Ritger–Peppas model is given by the following [56]:(7)$${\left[A\right]}_{t}={\mathrm{kt}}^{n}$$
where *k* is a release rate depending on the structural and geometrical characteristics of the release system, and *n* is the diffusional exponent defining the release mechanism.
(v)The Higuchi equation is expressed by the following equation [57].(8)$${\left[A\right]}_{t}=k\sqrt{t}\;$$
(vi)The Weibull model equation is given by the following [58]:(9)$$\log\left(-\ln\left(1-\frac{{\left[A\right]}_{t}}{100}\right)\right)=b\;\mathrm{logt}-\log a$$
where *a* is the scale parameter of the equation and determines the process time scale. *t* is the location parameter, which shows the lag time before the start of the release process (often zero). *b* is the shape parameter, which has three cases:Case 1: *b* = 1, an exponential curve;Case 2: *b* > 1, the release curve is s shaped or sigmoid with an upward curvature followed by a turning point;Case 3: *b* < 1, a parabolic curve with a high slope at initial step and then a consistent exponential decay curve.


#### 3.3.3. DPPH Radical-Scavenging Activity

DPPH evaluation was performed as previously reported [59] for nanoencapsulated CUR, DMC, and BDM. Therefore, the curcuminoid nanoparticle samples (BSA) were prepared in water. A 0.25 mM solution of 2,2-diphenyl-1-picrylhidrazyl (DPPH) was prepared in EtOH. Then, 0.5 mL of this solution was mixed with 1 mL of the respective nanoencapsulated curcuminoid solution at different concentrations. The solutions were incubated at 25 °C in the dark for 30 min, and DPPH absorbance was measured at 517 nm. Blanks were prepared for each concentration. The inhibition percentage was determined as shown in Equation (10).
(10)$$\mathrm{Inhibition}\;\mathrm{percentage}=\frac{\left(\mathrm{Absorbance}\;\mathrm{of}\;\mathrm{control}\;-\;\mathrm{Absorbance}\;\mathrm{of}\;\mathrm{sample}\right)}{\left(\mathrm{Absorbance}\;\mathrm{of}\;\mathrm{control}\right)}\times 100$$

The percentage of the radical-scavenging activity of the sample was plotted against its concentration to calculate IC_50_, which is the amount of sample required to reach 50% radical-scavenging activity. The samples were analyzed in three independent assays.

## 4. Conclusions

CUR-, DMC-, and BDM-loaded hybrid nanoparticles composed of BSA and cholesterol were successfully prepared by the high-speed homogenizer method and characterized by TEM, FT-IR, TGA, and DSC. The synthesized NPs showed desirable nanoparticles characteristics including particle size, monodisperse size distribution, shape, and encapsulation efficiency, therefore contributing to improving solubility, stability, and managing to successfully encapsulate the curcuminoids in the hybrid nanosystem. Furthermore, the BSA-based nanoparticles exhibited potential for drug release delivery in M_1_ and M_2_, showing considerable improvements in aqueous release. The release data were fitted using various kinetic models, and CUR and DMC showed to be concentration dependent in M_1_ in alignment with the presence of MeOH in it, whereas DMC in water suggested a release by diffusion. BDM fitted best with the model and suggested anomalous transport as the predominant mechanism of the release, which is indicative of agglomeration during diffusion.

The synthesized hybrid NPs were able to load CUR, DMC, and BDM efficiently with a stable release in in vitro profiles. DPPH radical scavenging activity indicated improvements in antioxidant activity. Therefore, the present work underlines the importance of natural bioactive molecules such as curcuminoids traditionally recognized for their nutritional and therapeutic benefits in their application to the pharmaceutical industry through nanoparticles for the purpose of effective encapsulation, protection, and release of such challenging bioactive molecules. 

## Figures and Tables

**Figure 1 molecules-27-02758-f001:**
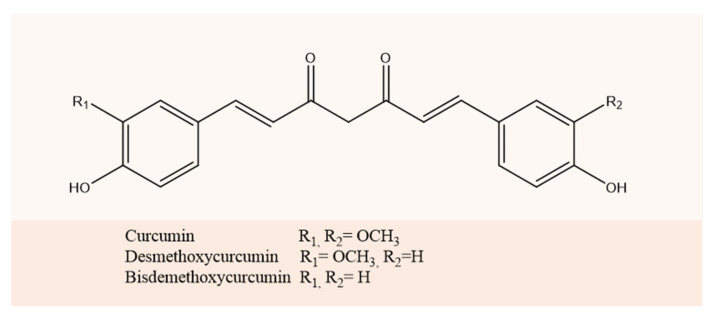
Chemical structure of curcuminoids.

**Figure 2 molecules-27-02758-f002:**
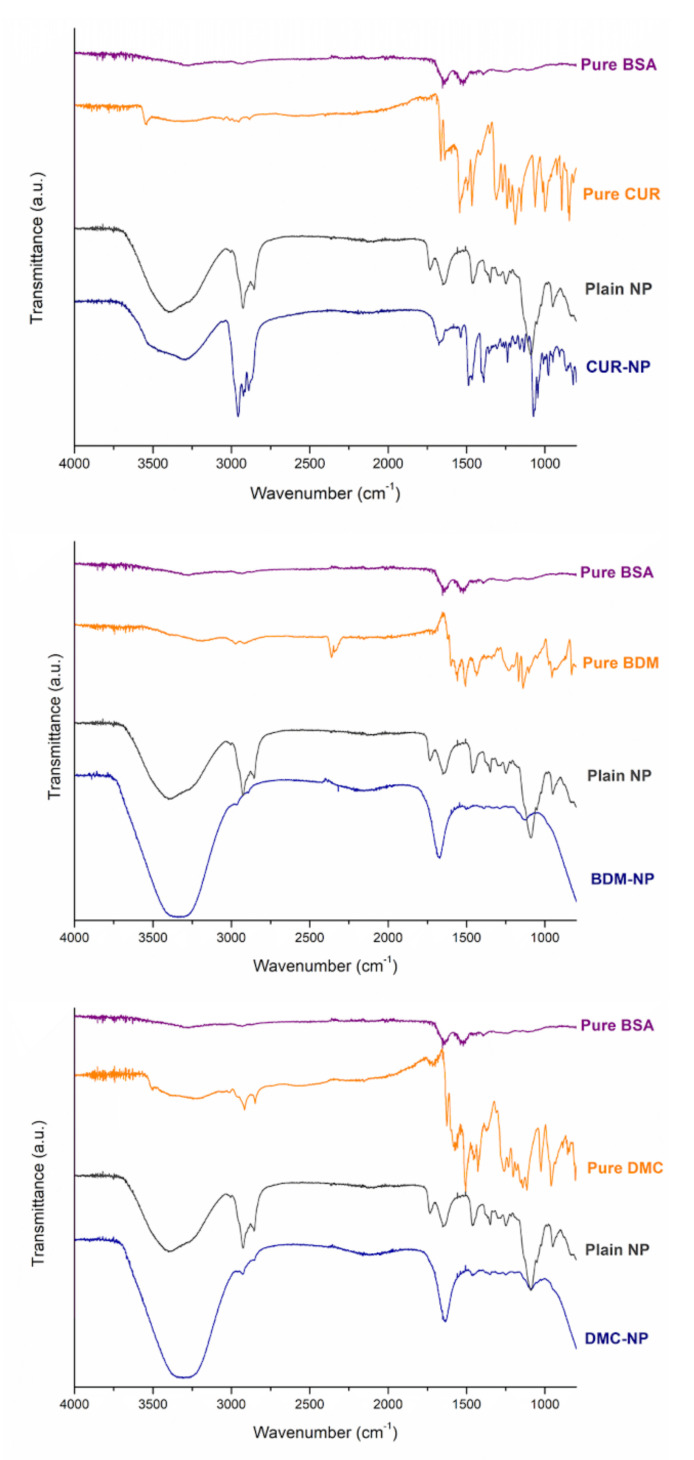
FT-IR spectra of CUR-, DMC-, and BDM-loaded BSA-NP, plain BSA-NP, and free CUR, DMC, and BDM.

**Figure 3 molecules-27-02758-f003:**
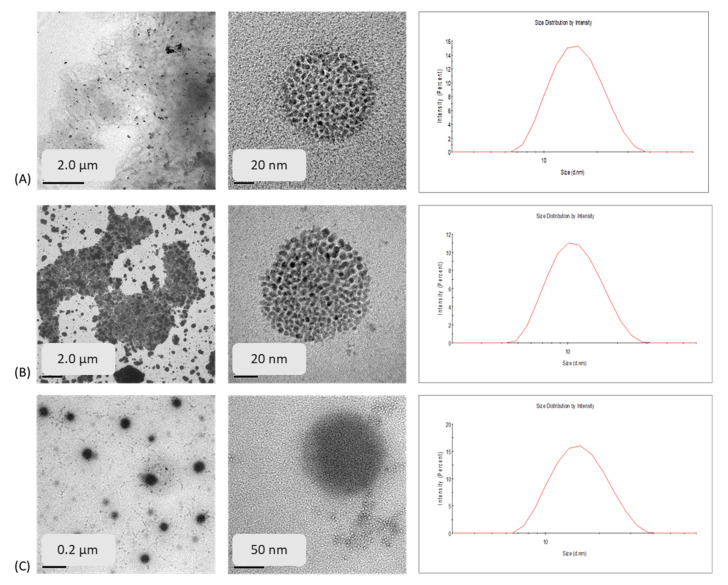
HR-TEM images and size distribution histogram of (**A**) CUR-NP, (**B**) DMC-NP, and (**C**) BDM-NP.

**Figure 4 molecules-27-02758-f004:**
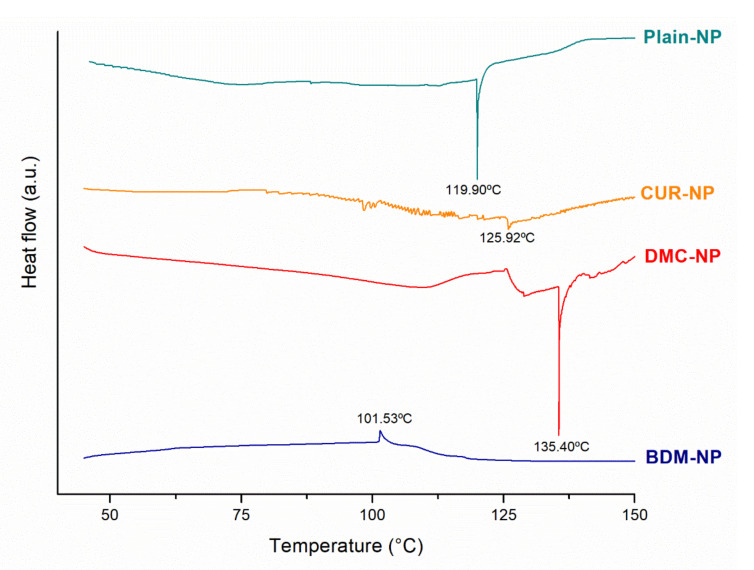
DSC curves of Plain NP, CUR-NP, DMC-NP, and BDM-NP formulations.

**Figure 5 molecules-27-02758-f005:**
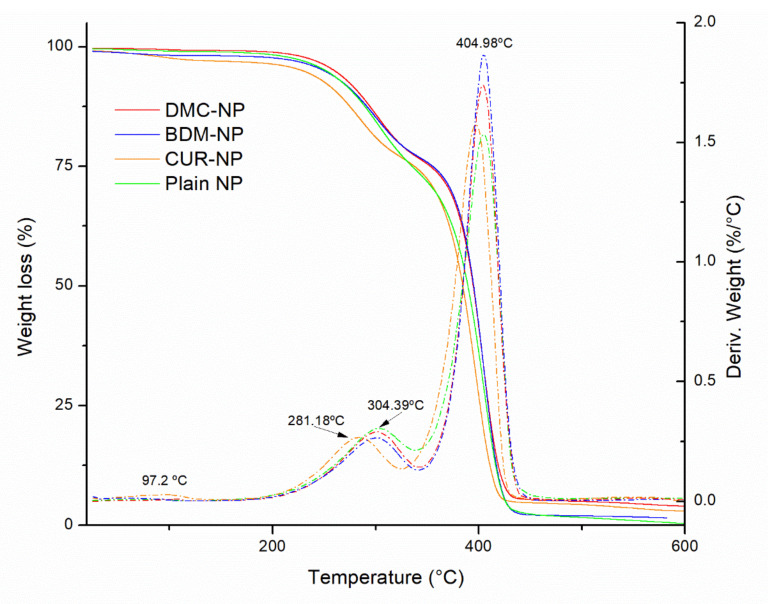
TGA curves of plain NP, CUR-NP, DMC-NP, and BDM-NP. Dashed and dotted lines represent the derivative of % weight loss as a function of temperature.

**Figure 6 molecules-27-02758-f006:**
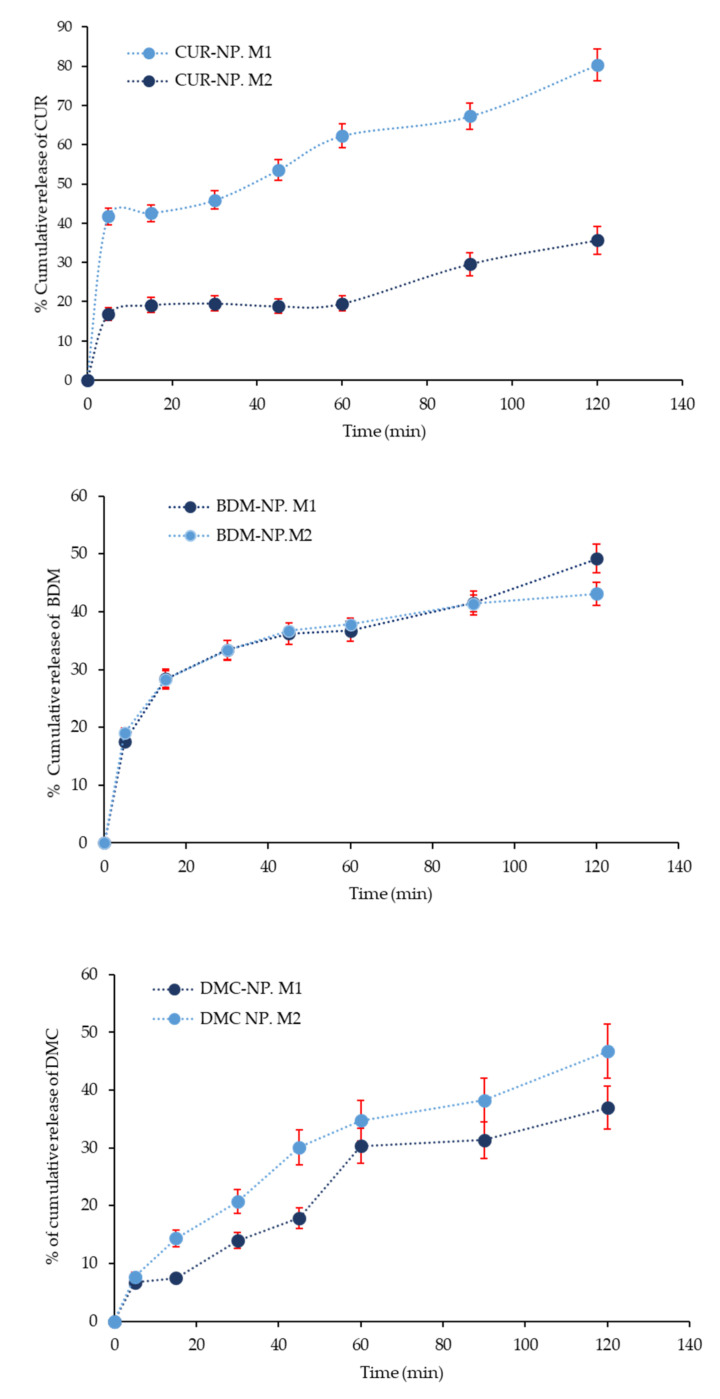
Release profile of CUR, DMC, and BDM from their respective hybrid NP formulation evaluated in two dissolution media M_1_ and M_2_. Error bars (red) represent the standard deviation of CUR, DMC, and BDM concentrations in triplicates.

**Figure 7 molecules-27-02758-f007:**
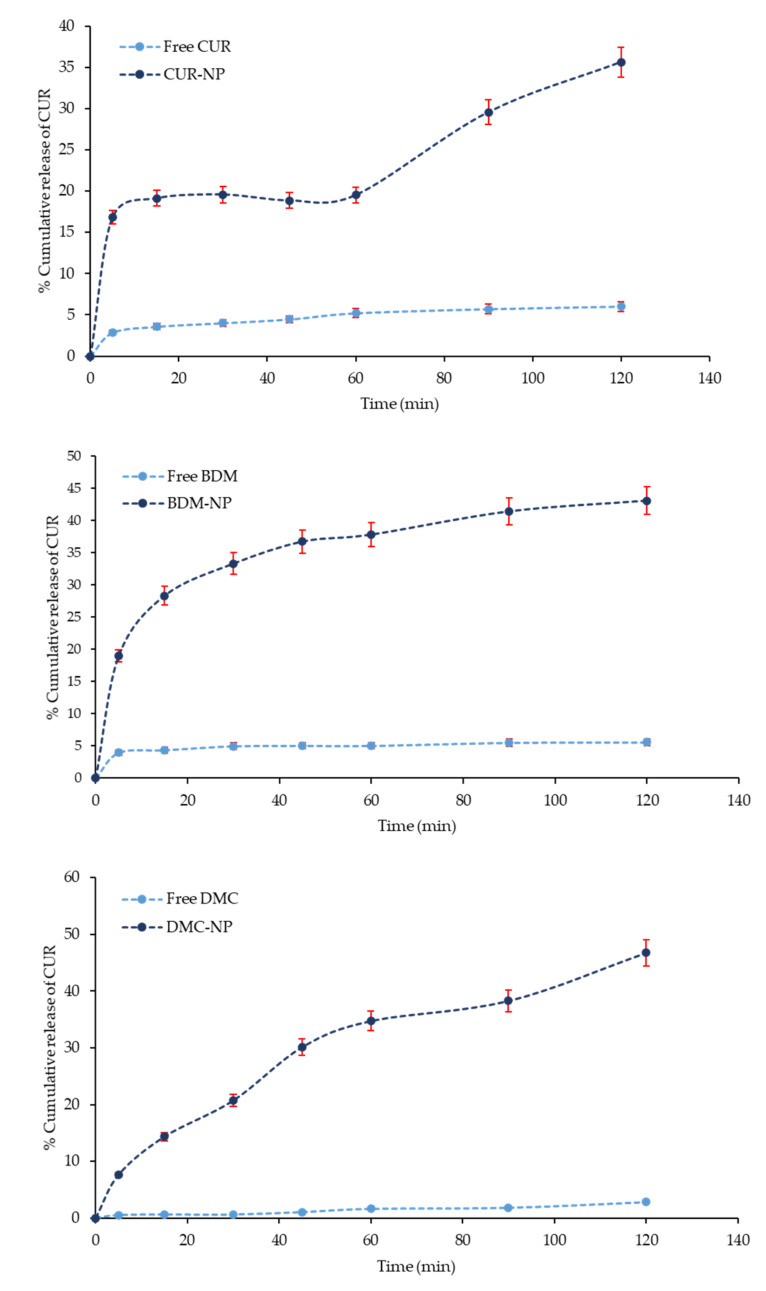
Release profile of CUR, DMC, and BDM from their respective hybrid NP formulation compared with the free compounds evaluated in water (M2). Error bars (red) represent the standard deviation of CUR, DMC, and BDM concentration in triplicates.

**Table 1 molecules-27-02758-t001:** Physical characteristics of CUR-, DMC-, and BDM-loaded NP.

Formulation	Size Average (nm) ^1^	PDI ^1^	EE (%) ^1^	LC (mg/g)	Yield (%)
CUR-NP	15.83 ± 0.18	0.250 ± 0.02	98.76 ± 0.68 ^a^	19.03 ± 0.45	93.35 ± 1.57
DMC-NP	17.29 ± 3.34	0.393 ± 0.05	98.99 ± 0.98 ^a^	28.34 ± 0.97	62.85 ± 1.52
BDM-NP	15.14 ± 0.14	0.156 ± 0.004	96.85 ± 0.26 ^b^	20.35 ± 0.94	85.68 ± 2.78

^1^ Values are expressed as mean ± S.D (*n* = 3). Different superscript letters in the same column indicate that differences are significant at *p* < 0.05 using ANOVA with a Tukey statistical test.

**Table 2 molecules-27-02758-t002:** Results of fitting the curcuminoids release kinetics experimental data to the kinetic equation proposed by six different kinetic models.

System	Zero Order			First Order			Second Order			Ritger-Peppas			Higuchi		Weibull		
	k	[A]_0_	R^2^	k	ln[A]_0_	R^2^	k	1/[A]_0_	R^2^	k	*n*	R^2^	k	R^2^	b	a	R^2^
CUR-NP M_1_	20.617	38.022	0.9862	0.356	3.689	0.9825	0.112	3.059	0.9778	32.756	0.290	0.9270	52.902	0.9595	0.176	1.109	0.9350
CUR-NP M_2_	9.569	14.709	0.9092	0.382	2.771	0.9274	0.001	−0.048	0.9133	28.147	0.245	0.8462	15.549	0.7924	0.072	3.830	0.6109
BDM-NP M_1_	10.781	24.886	0.7935	0.416	3.143	0.7541	−0.014	0.043	0.6129	18.293	0.560	0.9853	88.423	0.9542	0.148	2.059	0.9629
BDM-NP M_2_	13.585	22.909	0.8763	0.342	3.205	0.6946	−0.011	0.041	0.5924	17.378	0.613	0.9966	20.810	0.9477	0.138	2.045	0.9824
DMC-NP M_1_	2.750	21.266	0.9573	0.112	3.059	0.9778	0.005	−0.047	0.9610	4.4428	0.985	0.9135	4.1546	0.9320	0.027	3.634	0.8259
DMC-NP M_2_	3.192	20.799	0.9373	0.139	3.037	0.9262	0.006	−0.048	0.9080	4.3549	1.080	0.9032	4.9457	0.9700	0.034	3.621	0.9391

**Table 3 molecules-27-02758-t003:** Antioxidant activity of free and curcuminoids CUR-, DMC-, and BDM-loaded BSA-NP.

IC_50_ (µg/mL) ^1,2,3^			
	EtOH ^4^	Water ^4^	BSA
CUR	9.60 ^a,^* ± 0.12	2444.80 ^a,#^ ± 9.68	9.28 ^a,^* ± 0.29
DMC	12.46 ^b,^* ± 0.02	2143.07 ^b,#^ ± 0.61	11.70 ^b,^* ± 0.13
BDM	17.94 ^c,^* ± 0.06	1398.68 ^c,#^ ± 5.07	15.19 ^c,^* ± 0.57

^1^ IC_50_ µg curcuminoid/mL. ^2^ Values are expressed as mean ± standard deviation (S.D.). ^3^ Different superscript letters in the same column or different superscript signs in the same row indicate differences are significant (*p* < 0.05) using one-way analysis of variance (ANOVA) with a Tukey post hoc. ^4^ Results for DMC and BDM were previously reported by our group [50].

## Data Availability

Not applicable.

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
