# Peer review of "Bovine Serum Albumin-Based Nanoparticles: Preparation, Characterization, and Antioxidant Activity Enhancement of Three Main Curcuminoids from *Curcuma longa"

_molecules, 2022, doi:10.3390/molecules27092758_

Round 1
Reviewer 1 Report
The manuscript describes the preparation of BSA-based NPs loaded with three curcuminoids. The materials were characterized. Some experiments were also addressed to assess the ability of the materials to release the load in aqueous solutions and to evaluate antioxidant activity.
Generally, the paper might be found interesting for Molecules readers. However, I cannot recommend the publication of the manuscript in its current form. It is due to the poor English which has to be substantially improved before the manuscript resubmission (some examples have been indicated below). There is also some important information missing in the experiments description which is also unacceptable. Thus, I am asking authors to take an effort to improve the manuscript and reply to my comments.
The description in Section 4.2.4 is unclear. What metabolites were extracted? Equation [1] is also confusing (e.g. units?).
Section 2.2.1
In my opinion there is no significant difference in the IR spectra between plain NPs and loaded NPs to confirm the loading of phenols. If authors do not agree with my opinion, they are obligated to mark the characteristic bands on the loaded NPs spectra that prove the presence of phenols.
Section 2.2.2.
Was there any change in particles size after phenols loading? It would also be nice to provide information on the zeta potential before and after phenols loading.
Encapsulation efficiency parameter calculated by authors is dependent on the amount of curcuminoids added to the NPs. Besides the fact that the description in Section 4.2.4 unable critical review of this part of the work, the provided parameter is irrelevant. It would be much more informative to provide information about the ‘load-ability’ of NPs (how much curcuminoids can be loaded per e.g. 1 gram of NPs).
“Similar results have been demonstrated in previous studies also, where reduction in the 119 particle size of active ingredients to nanoparticle size has shown improvement in its efficacy, solubility, and bioavailability” – the sentence is confusing
“These parameters are important to help determine stability and the loaded function of the nanoparticles due to the influence in the release of the compound inside the nanoparticle” – the sentence is confusing. What do histograms have to do with stability and “loaded function” (whatever it means).
For me, the justification of the difference in the size of BDM-modified NPs observed with DLS and TEM is insufficient.
Section 2.3.1
In Section 4.3.1 there is no information about the amount of free curcuminoids used for comparison with modified NPs. How was the concentration (amount) of curcuminoids adjusted to experimental conditions?
“However, for these type 201 of molecules considering their water solubility its accepted dissolution media containing 202 surfactants or organic solvents.” – the sentence is completely confusing.
Section 2.3.3
Are the concentration values provided in Table 3 for BSA quantitated for BSA NPs concentration or the concentration of curcuminoids?
Author Response
Please find attach the Author's reply

Reviewer 2 Report
The manuscript entitled Bovine serum albumin-based nanoparticles: preparation, characterization, and antioxidant activity enhancement of three main curcuminoids from Curcuma longa is interesting in its conceptual design, but this needs to be matched with the quality of the manuscript. At present, sections about analysis methods and the writing need to be improved.
When comparing the sizes obtained by DLS for the three formulations, it was found similar sizes for the three of them. Revise this sentence.
- Some characterization techniques employed were unsuitable or unclear for such purposes Ex, FTIR - curcumin load.
- FTIR interpretation should be revised with suitable references.
The spacing between the numeral and the unit, i.e. 5.5%, to be corrected as 5.5 %,---- should be consistent throughout the manuscript.
Author Response
Please find attached the Author's reply

Reviewer 3 Report
General Comments:
This manuscript presented approach of designing the drug delivery system to overcome low bioavailability of poor solubility drugs such as curcuminoids using Bovine Serum Albumin (BSA). The three formulations: curcumin (CUR), desmetoxycurcumin (DMC) and bisdemethoxycurcumin (BDM) loaded BSA nanoparticles were prepared and characterized by the average size, size distribution, crystallinity, weight loss, drug release, kinetic mechanism and antioxidant activity. FT-IR analisys confirmed the encapsulation of curcuminoids and their spherical shape was confirmed by TEM. All formulations showed encapsulation efficiency upper to 96 %, increased antioxidant activity and release from the nanoparticles compared to free compounds in water suggesting proposed hybrid systems adequate for encapsulation, protection, and delivery of curcuminoids for the development offunctional foods and pharmaceuticsals. This topic should be of interest for the community of material and pharmaceutical science. Therefore, I suggest an acceptance for publication of this manuscript after the following issues have been well addressed.
Specific Comments:
- The references for the size average, polydispersity index and encapsulation efficiency of CUR, DMC and BDM loaded NP were missed.
- What was the rationale for the selection of the drug loading amount, and the applied concentration? Is that practically relevant? Some justification should be given in the text. The loading is relatively small.
Author Response
Please find attached the Author's reply

Round 2
Reviewer 1 Report
Most of my comments have been adequately addressed by the authors. However, the general quality of the language in the manuscript was not improved. Moreover, I have concerns on the correctness of the calculation of LC (loading capability) parameter.
I assume that LC was calculated with the results obtained for the calculation of EE (encapsulation efficiency). How is it possible to obtain almost complete encapsulation of all curcuminoids while the LC is differing so strongly? Moreover, I would prefer that LC parameter is given as mg/g or mM/g. Anyway, if 5 mg of each curcuminoid was mixed with 250 mg of BSA for loading and NPs synthesis, it is impossible to obtain >60% LC as it was stated by authors.
Author Response
Author's reply attached

Reviewer 2 Report
The authors have made the required changes. Hence I recommend the manuscript for publication.
Author Response
Thank you